# Effects of *Rheum palmatum* Root on In Vitro and In Vivo Methane Production and Rumen Fermentation Characteristics

**DOI:** 10.3390/ani14182637

**Published:** 2024-09-11

**Authors:** Seong Shin Lee, Jisoo Wi, Hyun Sang Kim, Pil Nam Seong, Sung Dae Lee, Jungeun Kim, Yookyung Lee

**Affiliations:** 1Animal Nutrition & Physiology Division, National Institute of Animal Science, Rural Development Administration, Wanju 55368, Republic of Korea; shin7398@korea.kr (S.S.L.);; 2Animal Products and Processing Division, National Institute of Animal Science, Wanju 55368, Republic of Korea; 3Department of Animal Sciences, The Ohio State University, Wooster, OH 44691, USA

**Keywords:** cattle, medicinal herb, methane, *Rheum palmatum*

## Abstract

**Simple Summary:**

This study was conducted to explore the optimal level of *Rheum palmatum* root (RP) for the mitigation of enteric methane and demonstrate the effects on methane production and rumen fermentation characteristics in cattle fed with RP. In the in vitro trial, 3% of RP addition most effectively reduced methane production, which determined the feeding level for the test subjects in the in vivo trial. In the in vivo trial, RP addition did not affect methane emissions in the steers, likely due to the adaptation of ruminal microorganisms to the RP addition.

**Abstract:**

This study investigated the impact of *Rheum palmatum* root (RP) for reducing methane and its impact on rumen fermentation and blood metabolites in cattle. Rumen fluid was collected from three cannulated steers (736 ± 15 kg) and mixed with buffer (1:3 ratio) for the in vitro trial. Treatments were divided into control and RP supplement groups (1%, 3%, and 5% of substrates), with each sample incubated at 39 °C for 24 and 48 hours. Methane was measured after incubation, showing a dose-dependent linear decrease after 48 hours. Quadratic changes were observed in total volatile fatty acids, acetate, and butyrate. Additionally, in vitro dry matter digestibility decreased linearly with RP inclusion. In vivo trials involved four Korean steers in a 2 × 2 crossover design over 3 weeks, with treatments including a control group and a group with 3% RP addition. Dry matter intake (DMI) tended to decrease in the RP group compared to the control. Methane emissions (g/kg DMI) were not affected by RP addition. Blood metabolites indicated higher lipase concentrations in the RP group. In conclusion, RP reduced methane production in the in vitro trial but had no effect in the in vivo trial, likely due to adaptation of ruminal bacteria to RP.

## 1. Introduction

The global average temperature has increased by 1.09 °C over the past decade [1], prompting many countries to acknowledge the severity of the climate crisis. Governments have established policies to achieve carbon neutrality, and the livestock industry is no exception [2]. In the beef and dairy cattle industry, methane is recognized as the primary greenhouse gas [3], and Glasson et al. [4] reported that its global warming potential is more than 28 times higher than that of carbon dioxide. Therefore, researchers are currently developing strategies to reduce methane emissions from ruminants. To regulate rumen fermentation characteristics and reduce greenhouse gas emissions using chemical compounds such as antibiotics, nitrates, and sulfate, various methods have been explored [5]. However, antibiotics are prohibited in many countries, and certain chemical compounds still present various issues, such as the presence of residues in beef and increasing resistance to antibiotics [6].

In contrast, medicinal herbs which have been used for various purposes such as food and pharmaceuticals for a long time, have been recognized for their potential as stable methane inhibitors [7]. Several studies have shown that medicinal herbs regulate rumen fermentation characteristics, improve nutrient utilization, and reduce methane emissions from ruminants [8]. Moreover, when used alone or in combination, medicinal herbs have been reported to enhance production and rumen fermentation efficiency, while reducing methane emissions in terms of microbial cell synthesis and ruminal volatile fatty acid (VFA) production [9].

The root of *Rheum palmatum* (also known as rhubarb) was used by ancient Chinese people to treat digestive disorders and was demonstrated to be a stable additive [10]. In a recent study, *Rheum palmatum* was shown to have antibacterial, hypocholesterolemic, and antiseptic effects [11]. The main phytochemicals in *Rheum palmatum* are emodin, physcion, aloe-emodin, rhein, chrysophanol, chrysophanol glycosides, emodin glycosides, and anthraquinone [12]. In ruminants, these phytochemicals improve rumen fermentation and decrease methane production [13]. Several studies have demonstrated the effects of *Rheum palmatum* on the improvement of rumen fermentation in an in vitro batch system; however, the feeding experiments of *Rheum palmatum* were limited [14]. Therefore, this study was conducted to explore the optimal level of *Rheum palmatum* root for the mitigation of enteric methane production and demonstrate the effects of methane reduction in cattle fed with *Rheum palmatum* root additive.

## 2. Materials and Methods

### 2.1. Herb (Rheum palmatum) preparation and composition

The root of *Rheum palmatum* was purchased from a local market (Junju, Republic of Korea). Chemical analyses of RP were performed using AOAC methods [15]: crude protein (CP), #942.05; ether extract, #920.39; and crude ash, #954.01. The neutral detergent fiber (NDF) and acid detergent fiber (ADF) contents were analyzed using an ANKOM2000 fiber analyzer (ANKOM Technology Corporation, Macedon, NY, USA), based on Van Soest et al. [16]. The CP content was calculated as nitrogen × 6.25. Non-fiber carbohydrate (NFC) was calculated as 100 − (CP + ether extract + crude ash + NDF). The major phytochemical compounds in *Rheum palmatum* root were analyzed by EZmass (Jinju, Republic of Korea) using a UPLC-Q-TOF MS system (Waters, Milford, MA, USA). EZmass Company’s main phytochemical compound analysis conditions were as follows: metabolites were separated using an Acquity UPLC BEH C_18_ column (2.1 mm × 100 mm, 1.7 m; Waters Corp., Milford, MA, USA) equilibrated with water containing 0.1% formic acid (solvent A) and eluted with a gradient of acetonitrile containing 0.1% formic acid (solvent B). Compounds were detected from *m*/*z* 50–1500 with a scan time of 0.2 s using the Q-TOF MS system in positive or negative mode using the following instrument settings: capillary voltage, 3.0/2.5 kV; sample cone voltage, 30/20 V; ion source temperature, 100 °C; desolvation temperature, 400 °C; cone gas flow rate, 30 L/h; and desolvation gas flow rate, 800/900 L/h. All analyses were performed using a lock spray to ensure accuracy and reproducibility. Leucine-enkephalin ([M + H] = 556.2771 Da; [M − H] = 554.2615 Da), used as lock mass, was infused at a flow rate of 20 μL/min and a frequency of 10 s. MS/MS spectra were obtained using a collision energy ramp from 10 to 30 eV. LC-MS data, including retention time, *m*/*z*, and ion intensity, were extracted using UNIFI version 1.9.2.045 (Waters) and assembled into a data matrix. The compounds were tentatively identified using online databases, including the ChemSpider database, Metlin database (www.metlin.scripps.edu (accessed on 10 August 2021)), human metabolome database (www.hmdb.ca (accessed on 10 August 2021)), EZmass database, and authentic standards. The chemical composition of *Rheum palmatum* root is listed in Table 1.

### 2.2. In Vitro Experiment

#### 2.2.1. Animals and Diet Composition

Three rumen-cannulated Korean native Hanwoo steers (39 months old), averaged 736 ± 15 kg of body weight (BW), were used as donor animals. Steers were fed diets consisting of 9.4 kg/d concentrate and 2.4 kg/d mixed hay (as-fed basis) at 1000 and 1500 h. The animals had ad libitum access to water and mineral blocks. Ingredients and chemical composition of the experimental diets are presented in Table 2. To determine dry matter (DM) content, feed samples were dried in a drying oven at 60 °C for 48 h, then milled in a Cyclotec 1093 meal (FOSS, Suzhou, China) through a 1 mm screen. The prepared samples were stored at −20 °C until further analysis. Chemical analysis was conducted using the AOAC method [15] as previously described.

#### 2.2.2. Incubation Conditions

One hour before the morning feed, an equal volume of ruminal fluid was collected from the three Hanwoo steers. The fluids were pooled, mixed in a flask, and filtered through four layers of cheesecloth. The rumen fluid was immediately transported to the laboratory for in vitro incubation. Substrates consisted of 0.4 g of concentrate and 0.1 g of mixed hay (DM basis), which were fed to donor animals. The ruminal fluid was mixed with McDougall buffer [17] containing L-cysteine (0.5 g) at a ratio of 1:2 (*v*:*v*), and the buffered rumen fluid was flushed by carbon dioxide gas. The substrate (0.4 g concentrate and 0.1 g mixed hay) was weighed and placed in an Ankom F57 filter bag (Ankom Technology Corp., Macedon, NY, USA). The 1%, 3%, and 5% *Rheum palmatum* root powder treatment groups were supplemented with the respective addition levels in each filter bag. Then, 50 mL of buffered rumen fluid and the Ankom filter bag were dispensed into a 125 mL serum bottle. Carbon dioxide gas was flushed into the headspace of the serum bottle, then serum bottles were immediately incubated in a shaking incubator (39 °C, 120 rpm) for 24 and 48 h. The treatments were performed in quadruplicate, and the experimental unit was a serum bottle (24-h and 48-h incubation bottles are independent).

#### 2.2.3. Sample Collection and Analysis

After 24 or 48 h of incubation, the produced gas in each serum bottle was collected in a gas-tight aluminum bag (85 mm × 135 mm, 9 Ø, 200 mL) with a silicon sealer (9–11 Ø, 18 mm) using a 150 mL glass syringe. The collected gas was analyzed using a gas chromatograph (GC; NL/450 GC, Bruker, Billerica, MA, USA) equipped with a capillary column (GS-GASPRO 113-4332, 30 m × 0.320 mm, Agilent Technologies Inc., Santa Clara, CA, USA) to measure methane production. The pH of the incubated fluid was measured using a pH meter (SevenEasy pH; Mettler-Toledo AG, Schwerzenbach, Switzerland). The residual rumen fluid was transported to a new tube for analysis of volatile fatty acids (VFAs), ammonia nitrogen (NH_3_-N), and rumen microbes. Thereafter, samples for microbial analysis were stored in liquid nitrogen until further analysis. The substrates were dried in a drying oven at 60 °C for 48 h to measure the digestibility. The in vitro DM digestibility (IVDMD) was calculated using the dried weight of the substrate samples.

The concentrations of VFAs and NH_3_-N were analyzed using the methods described by Erwin et al. [18] and Chaney and Marbach [19], with minor modifications. Briefly, five milliliters of supernatants after centrifugation of rumen fluid was mixed with 500 μL of 50% metaphosphoric acid (MPA; Catalog number 239275, Sigma-Aldrich, St. Louis, MP, USA) for VFAs or 500 μL of 25% MPA for NH_3_-N, then mixtures were stored at −80 °C until analysis. For analysis of VFAs, the fluid mixed with MPA was centrifuged at 14,000× *g* for 10 min at 4 °C. One milliliter of supernatant was distributed to a GC analysis vial, and samples were analyzed by GC (6890N, Agilent Technologies, Wilmington, DE, USA) with a capillary column (Nukol™ Fused silica capillary column, 15 m × 0.53 mm × 0.5 µm, Supelco Inc, Bellefonte, PA, USA). A standard curve was generated using the Volatile Free Acid Mix (catalog number. CRM46975; Sigma-Aldrich). For NH_3_-N analysis, the fluid mixed with MPA was centrifuged at 14,000× *g* for 5 min at 4 °C. After centrifugation, twenty microliters of supernatants was mixed with 1 mL of phenol color reagent (50 g/L of phenol plus 0.25 g/L of nitroferricyanide) and 1 mL of alkali-hypochlorite reagent (25 g/L of sodium hydroxide and 16.8 mL/L of 4–6% sodium hypochlorite). The mixture was colored in a 37 °C water bath for 15 min. After coloring, 8 mL of distilled water was added, and the NH_3_-N concentration was determined by measuring the absorbance at 630 nm using a UV spectrophotometer (Bio-Rad, US/benchmark plus, Tokyo, Japan).

### 2.3. In Vivo Experiment

#### 2.3.1. Respiratory Chamber Conditions and Methane Gas Measurement

Methane emissions in this experiment were measured using four open-circuit respiratory chambers. The chambers were operated and calibrated as described by Pinares-Patiño et al. [20]. The size of the chamber facility size was 3250 mm × 4750 mm × 2500 mm. The chamber had walls and a door made of stainless steel (SUS304), which were filled with urethane form (thermal conductivity: 0.0185 kcal/mh°C) and double-pane windows containing argon gas (thermal conductivity: 0.0094 kcal/mh°C) used as heat insulators. In addition, a metabolic cage made of steel pipes (1400 mm × 2950 mm × 2120 mm) was fixed within the chamber to keep the animals in one place. Supply and exhaust air ducts were installed at the front and back sides of the chamber, respectively, and nonwoven prefilters were installed at each air duct to prevent penetration of dust or animal hair. Four 24 V air pumps (DMC-DAP-3657B DC24V, Motorbank, Seoul, Republic of Korea) were installed in the exhaust duct to sample the air. Air samples were vented through an infrared methane sensor (Airwell plus, Kinsco technology, Seoul, Republic of Korea) to measure methane emissions within a 0–200 ppm detectable range (±0.2 ppm resolution). Environmental conditions within the chamber, including temperature, humidity, airflow, and pressure, were measured using auto-sensors (EE071, 671, 610; Eplus, Engerwitzdorf, Austria) for more precise methane analysis. The internal temperature of the chamber was maintained at 22 °C, relative humidity was at 60%, and airflow was 2800 L/min. The recovery rate of methane gas was 92.4 ± 5.38%, and it was used to calculate the measured methane yields from the animals.

#### 2.3.2. Animals, Diet Composition, and Experimental Design

Four Korean native steers (Hanwoo, 393 ± 33.5 kg of BW, 27 months old) were randomly assigned to one of 2 treatments in a 2 × 2 crossover design with a 3-week period. Each animal was placed in an open-circuit respiratory chamber. The steers stayed in a metabolic cage, which was fixed in the chamber, and had free access to water. Treatments were as follows: diets (concentrate/forage = 4:1) with no additives (CON) and diets supplemented with 3% *Rheum palmatum* root based on dry matter (RP). Steers were fed diets based on 2% of their initial BW during each period. Equal amounts of diet were provided to the steers at 1000 and 1500 h. The ingredients of the concentrate were the same as those used for the in vitro experiments (Table 2). The RP was added when the concentrate was pelleted. Chemical analysis of the diets was conducted using AOAC methods [15], as previously described. The chemical compositions of the experimental diets are presented in Table 3. Steers were adapted to the chamber and diets for 2 weeks, and methane emissions and feed intake were measured for 1 week. For a day, methane gas was measured every 5 min in the chamber. The equation for methane yield calculation [20] is as follows: methane emission (L/min) = {(Dry standard temperature and pressure ventilation rate) × Methane (ppm) ÷ 1,000,000} ÷ (Methane recovery rate). Feed intake was measured by calculating the difference between the provided feed and residual feed (DM basis). The washout period was conducted for 14 days independently with a 21-day experimental period.

#### 2.3.3. Sample Collection and Analysis

On the last day of each period, blood and ruminal fluid were collected 4 h after morning feeding. Ten milliliters of blood samples was obtained from the jugular vein of steers and collected into EDTA tubes (BD Vacutainer^®^, Eysins, Switzerland) for serum collection. Blood samples were centrifuged at 2000× *g* for 15 min at 4 °C. Subsequently, the supernatants were transported to new tubes and stored at −20 °C until analysis. Serum samples were analyzed for glucose, creatinine, blood urea nitrogen, inorganic phosphate, calcium, total protein, albumin, globulin, alanine aminotransferase, alkaline phosphatase, gamma-glutamyl transferase, total bilirubin, cholesterol, amylase, and lipase levels using an IDEXX Catalyst DX system (IDEXX Laboratories Inc., Cary, NC, USA). Two hundred milliliters of ruminal fluid was collected by a stomach tube comprising the head part (length 13 cm, diameter 3 cm) and flexible tube (length 120 cm, diameter 1 cm) with a vacuum pump (Welch and Thomas, Mt. Prospect, IL, USA). The pH of ruminal fluids, VFAs, and NH_3_-N were analyzed using the methods described in the in vitro experiment section.

### 2.4. Statistical Analysis

All data from the in vitro experiments were analyzed using the GLM procedure in SAS (Enterprise Guide 7.1, SAS Institute Inc., Cary, NC, USA). The incubation time and RP treatments were assigned to the statistical model, and fixed effects were considered. Differences between the treatment means were analyzed using Tukey’s multiple comparison test. In the in vitro experiment, orthogonal polynomials were used to test the linear, quadratic, and cubic effects of RP addition on changes in ruminal variables. All data are presented as least-squares means, and significance was declared at *p* < 0.05.

All data from the in vivo experiments were analyzed using the GLIMMIX procedure of SAS (Enterprise Guide 7.1, SAS Institute Inc., USA). Treatment, period, and steer were assigned to a statistical model. Treatment was considered a fixed effect, and period and steer were considered random effects. Differences between treatment means were analyzed using Tukey’s multiple comparison test. All data are presented as least-squares means. Significance was declared at *p* < 0.05 and tendency was determined at 0.05 ≤ *p* < 0.10.

## 3. Results

### 3.1. Composition of Rheum palmatum Root

The chemical composition and 10 major components of the RP are listed in Table 1. RP showed a high level of NFC (63.75% on a DM basis) and a low level of fiber (NDF, 11.80%). The major components of RP accounted for 37.33% of all the phytochemicals detected. These chemicals mainly consist of stilbenoids (piceatannol, rhapontigenin, and their derivatives) and anthraquinones (emodin, chrysophanol, and their derivatives).

### 3.2. In Vitro Experiment: Ruminal Fermentation Characteristics

Fewer total VFAs were produced in the 5% RP group, with a quadratic decrease (*p* < 0.001; Table 4) compared to the other groups (*p* < 0.005). The proportion of acetate was higher in the 1% RP group than in the 3% RP group (*p* = 0.043). Butyrate increased linearly (*p* = 0.011) and iso-butyrate showed a quadratic increase (*p* < 0.001) with increasing levels of RP. The RP addition did not affect ruminal pH, total and methane gas production, NH_3_-N concentration, propionate, valerate, isovalerate, acetate/propionate ratio (A/P ratio), or IVDMD after 24 h of incubation (*p* > 0.05).

After 48 h of incubation, methane yield (mL; *p* = 0.002), DM-adjusted methane (mL/g of DM; *p* = 0.002), and percentage of methane in the total gas (%; *p* = 0.017) decreased linearly in a dose-dependent manner. Quadratic changes were observed for total volatile fatty acids (*p* = 0.001), acetate (*p* = 0.029), butyrate (*p* < 0.001), and valerate (*p* = 0.001). The IVDMD decreased linearly with RP inclusion (*p* = 0.023). The RP addition did not change the pH, total gas production, digestible DM (dDM)-adjusted methane (mL/g of dDM), NH_3_-N, or A/P ratio (*p* > 0.05) (Table 5).

### 3.3. In Vivo Experiment

#### 3.3.1. Dry Matter Intake and Methane Emission

Table 6 shows the dry matter intake and methane emission of Hanwoo cattle. Dry matter intake (DMI) tended to decrease in the RP group compared to the CON group (*p <* 0.05). However, methane emissions (g/kg DMI) were not affected in the RP group compared with the CON group (*p* > 0.05).

#### 3.3.2. Rumen Fermentation Characteristics

Table 7 shows the rumen fermentation characteristics of Hanwoo cattle. None of the contents were affected by *Rheum palmatum* root addition, except for acetate and iso-butyrate concentrations. Acetate concentration was higher in the RP group than in the CON group (*p* < 0.05). The concentration of iso-butyrate was higher in the CON group than in the RP group (*p* < 0.05). The concentration of iso-valerate tended to decrease in RP group (0.05 ≤ *p* < 0.10).

#### 3.3.3. Blood Metabolites

Table 8 shows the blood metabolites in Hanwoo cattle. There were no differences in blood metabolites between treatments, except for the concentration of lipase. The lipase concentration was higher in the RP group than in the CON group (*p* < 0.05).

## 4. Discussion

### 4.1. In Vitro Experiment

In this study, the main chemical components of *Rheum palmatum* root were stilbenoids (piceatannol, rhapontigenin, and their derivatives) and anthraquinones (emodin, chrysophanol, and their derivatives). Although the chemical composition of *Rheum palmatum* root in the study differed from that of other studies [21,22], the major chemical components were similar to those of previous studies. Anthraquinones, among the main chemical components of *Rheum palmatum* root, are known to exert antibacterial effects by inhibiting bacterial protein synthesis and membranes [11]. García-González et al. [23] reported that the anthraquinones of *Rheum palmatum* are the main molecules that inhibit methanogenic bacteria, leading to a decrease in enteric methane production. This study showed similar results of a linear decrease in methane production (mL/g dDM) with increased levels of *Rheum palmatum* root addition.

The in vitro DM digestibility decreased with above 3% *Rheum palmatum* root addition (Table 4), and the results were similar to a study by Wang et al. [24], which included 2.8% *Rheum palmatum* addition. In addition, Vázquez-Carrillo et al. [9] reported that 3% addition of medicinal herbs decreased the digestibility of nutrients in ruminants by inhibiting fibrolytic bacteria. Our study showed results similar to those of previous studies, and it was expected that *Rheum* spp. would have a negative effect on rumen microorganisms, such as fibrolytic bacteria, due to its antibacterial activity.

García-González et al. [23] reported that the concentration of butyrate increased linearly in buffered rumen fluid incubated for 24 h, and expected that *Rheum* spp. could inhibit methanogenic archaea and lead to the accumulation of hydrogen in the rumen fluid. The accumulated hydrogen may have enhanced the oxidation of NADH, leading to an increase in the concentration of butyrate [25]. This study also showed that butyrate concentrations increased with *Rheum* spp. addition in buffered rumen fluid incubated for 24 h. This may have been due to the accumulation of hydrogen resulting from the inhibition of methanogenic archaea. The acetate-to-propionate ratio decreased linearly in buffered rumen fluid incubated for 48 h with an increase in propionate concentration. A decrease in the acetate-to-propionate ratio with all additives is associated with a decrease in gas production [25]. García-González et al. [23] reported that methanogenic archaea could be inhibited in the rumen by *Rheum* spp., leading to decreased methane production, acetate-to-propionate ratio, and H_2_ recovery. The results of that previous study were similar to those of this study as both studies showed a decrease in methane production and acetate-to-propionate ratio by increasing the dose of *Rheum palmatum*. Generally, IVDMD and VFAs have a positive correlation, but the present study showed different results (Table 5). Huang et al. [26] reported that gallotannins produced by plants can be utilized by certain rumen microorganisms to produce VFAs and CO_2_. Similarly, Buccioni et al. [27] reported an increase in total VFAs upon the addition of gallotannin extracts from chestnuts. Rhubarb, a medicinal plant known to be rich in gallotannins [28], is also predicted to lead to increased VFA production in this study. However, further research is needed to confirm these findings.

### 4.2. In Vivo Experiment

Akbar [29] reported that *Rheum* spp. have a bitter and distinctive smell that can decrease palatability. A previous study reported a decrease in DMI when animals were fed with a diet mixed with *Rheum palmatum* [9]. In this study, the diet mixed with *Rheum palmatum* root also showed low palatability compared to the control diet and tended to decrease DMI. According to Benchaar et al. [30], some phytochemicals produced by plants cannot mitigate enteric methane in ruminants, even though they effectively reduce methane in in vitro experiments. The adaptation of ruminal bacteria could be one of the reasons for the lack of enteric methane reduction [31]. In this study, the adaptation period was 14 d, which was expected to have no methane reduction effect owing to the herbal adaptation of rumen bacteria.

Several studies [32,33] reported that phenolic compounds produced by plants can reduce protein digestibility and branched-chain fatty acids (BCFA). This study showed similar results, but ammonia-N concentration was not changed. According to Yao et al. [34], BCFA and ammonia are both products of protein fermentation and typically exhibit a positive correlation. However, in the present study, while there was no change in ammonia nitrogen levels, BCFA levels decreased. A similar result was observed in the study by Burgos-Edwards et al. [35], where plant extracts inhibited protein fermentation, resulting in a decrease in BCFA levels, but no change in ammonia levels. They hypothesized that this was because ammonia could be formed from urea.

Previous studies have reported that the normal range of blood lipase levels in cattle is between 25 and 83 U/L [36,37], and the results of the current study fall within this range. In general, *Rheum palmatum* root is known as an inhibitor of lipase [38]. The concentration of lipase was higher in the RP group than in the CON group (*p* = 0.029; 57.0 vs. 36.7 U/L) in this study. Lee et al. [39] reported that lipase levels increase to maintain sufficient body energy due to restricted energy intake or body weight loss. It is expected that the change in lipase in the present study may negatively affect body weight gain in cows fed *Rheum palmatum* root.

## 5. Conclusions

In conclusion, *Rheum palmatum* root addition reduced methane production in an in vitro batch culture system, but had no effect in the feeding trial. This discrepancy may be attributed to adaptation of ruminal bacteria to *Rheum palmatum* root. Further studies analyzing ruminal bacterial communities are required to verify this hypothesis.

## Figures and Tables

**Table 1 animals-14-02637-t001:** Chemical composition of *Rheum palmatum* and major components of phytochemicals.

Item	*Rheum palmatum* Root
Chemical composition, % of dry matter	
Crude protein	8.31
Ether extract	0.99
Neutral detergent fiber	11.80
Acid detergent fiber	10.25
Crude ash	15.15
Non-fiber carbohydrates ^1^	63.75
**Phytochemicals ^2^**	**% of total**
Piceatannol-glucoside 1	6.53
Rhapontigenin-galloylglucoside 2	5.88
Piceatannol	5.60
Rhapontigenin-galloylglucoside 1	5.04
Rhapontigenin	3.45
Emodin	2.67
Chrysophanic acid	2.64
Isorhapontin	2.13
Torachrysone-8-glucopyranoside	1.82
Deoxyrhapontigenin-galloylglucoside 2	1.57

^1^ Non-fiber carbohydrates = 100 − (crude protein + ether extract + crude ash + neutral detergent fiber). ^2^ The top ten phytochemicals are presented.

**Table 2 animals-14-02637-t002:** Ingredients and chemical composition of in vitro experimental diets.

Item	Concentrate	Mixed Hay
Ingredients, % of dry matter		
Cracked corn	58.10	
Soybean hull	16.60	
Corn gluten feed	11.70	
Wheat bran	5.80	
Soybean meal	3.50	
Lupin	2.30	
Limestone	0.80	
Sodium bicarbonate	0.60	
Salt	0.40	
Vitamin and mineral mixture ^1^	0.20	
Kentucky bluegrass		50.00
Tall fescue		50.00
Chemical composition, % of dry matter		
Crude protein	12.79	14.21
Ether extract	3.09	0.94
Neutral detergent fiber	21.60	11.66
Acid detergent fiber	11.66	35.79
Crude ash	3.98	6.57
Non-fiber carbohydrates ^2^	58.55	6.99

^1^ Vitamin and mineral mixture: vitamin A, 2,650,000 IU; vitamin D3, 530,000 IU; vitamin E, 1050 IU; nicotinic acid, 10,000 mg; iron, 13,200 mg; manganese, 4400 mg; zinc, 4400 mg; copper, 2200 mg; iodine, 440 mg; cobalt, 440 mg. ^2^ Non-fiber carbohydrates = 100 − (crude protein + ether extract + crude ash + neutral detergent fiber).

**Table 3 animals-14-02637-t003:** Chemical composition of in vivo experimental diets.

Item	Control ^1^	RP ^1^	Italian Ryegrass Hay
Dry matter (DM), %	97.80	88.70	83.50
Crude protein, % of DM	19.68	18.40	9.63
Ether extract, % of DM	3.80	3.64	1.53
Neutral detergent fiber, % of DM	28.13	24.90	59.90
Acid detergent fiber, % of DM	15.22	14.10	37.20
Crude ash, % of DM	5.99	5.50	3.91
Non-fiber carbohydrate ^2^, % of DM	42.40	47.50	25.03

^1^ Control, no additives; RP, concentrate supplemented with 3% *Rheum palmatum* root on a dry matter basis. ^2^ Non-fiber carbohydrates = 100 − (crude protein + ether extract + crude ash + neutral detergent fiber).

**Table 4 animals-14-02637-t004:** Effects of *Rheum palmatum* root addition on in vitro gas production, ruminal fermentation characteristics, and digestibility after 24 h.

Variable ^1^	*Rheum palmatum* Root ^2^				SEM	*p*-Value		
	0%	1%	3%	5%		L	Q	C
pH	6.54	6.50	6.49	6.54	0.01	0.908	0.222	0.792
Total gas, mL	81.00	80.33	84.33	79.25	1.10	0.924	0.503	0.585
CH_4_ production								
mL	4.79	4.60	4.68	4.19	0.13	0.177	0.605	0.552
mL/g of DM	9.57	9.18	9.35	8.37	0.26	0.176	0.608	0.553
mL/g of dDM	15.92	15.41	15.17	13.90	0.43	0.036	0.666	0.643
%	5.92	5.73	5.43	5.26	0.15	0.010	0.651	0.987
NH_3_-N, mg/dL	17.21	17.95	18.51	18.40	0.30	0.234	0.448	0.907
Total VFAs, m*M*	91.80 ^a^	95.46 ^a^	99.75 ^a^	81.13 ^b^	3.98	0.010	<0.001	0.254
Acetate, %	54.65 ^ab^	56.05 ^a^	53.90 ^b^	54.55 ^ab^	0.45	0.178	0.888	0.011
Propionate, %	26.25	26.51	26.96	25.61	0.28	0.530	0.199	0.687
Butyrate, %	14.53 ^ab^	13.43 ^b^	14.92 ^a^	15.34 ^a^	0.41	0.011	0.233	0.020
Iso-butyrate, %	1.02 ^b^	0.88 ^b^	0.88 ^b^	1.24 ^a^	0.09	0.003	<0.001	0.893
Valerate, %	1.54	1.38	1.61	1.57	0.05	0.184	0.850	0.039
Iso-valerate, %	2.00	1.75	1.73	1.68	0.07	0.056	0.270	0.327
A/P ratio	2.09	2.12	2.00	2.13	0.03	0.944	0.308	0.263
IVDMD, %	61.65	60.65	61.65	59.88	0.35	1.000	0.422	0.820

^a, b^ Means without a common superscript letter differ significantly (*p* < 0.05). L, Q, and, C mean linear, quadratic, and cubic patterns, respectively. ^1^ DM = dry matter; dDM, digestible dry matter; VFAs, volatile fatty acids; A/P ratio, acetate/propionate ratio; IVDMD, in vitro dry matter digestibility. ^2^ Substrates containing 0%, 1%, 3%, and 5% of *Rheum palmatum* root on a dry matter basis.

**Table 5 animals-14-02637-t005:** Effects of *Rheum palmatum* root addition on in vitro gas production, ruminal fermentation characteristics, and digestibility after 48 h.

Variable ^1^	*Rheum palmatum* Root ^2^				SEM	*p*-Value		
	0%	1%	3%	5%		L	Q	C
pH	6.37	6.37	6.35	6.35	0.01	0.251	0.805	0.611
Total gas, mL	117.25	114.25	114.50	120.00	1.35	0.451	0.222	0.883
CH_4_ production								
mL	8.44 ^a^	7.77 ^ab^	7.40 ^b^	7.31 ^b^	0.26	0.002	0.090	0.466
mL/g of DM	16.85 ^a^	15.52 ^ab^	14.78 ^b^	14.61 ^b^	0.51	0.002	0.094	0.475
mL/g of dDM	23.14	22.09	21.29	20.97	0.48	0.017	0.351	0.744
%	7.20 ^a^	6.80 ^b^	6.46 ^b^	6.10 ^c^	0.24	<0.001	0.213	0.272
NH_3_-N, mg/dL	26.49	28.81	25.59	27.44	0.69	0.862	0.784	0.123
Total VFAs, m*M*	111.65 ^c^	139.53 ^a^	136.56 ^ab^	116.76 ^bc^	6.99	0.909	0.001	0.074
Acetate, %	53.54 ^b^	55.73 ^a^	54.00 ^ab^	52.87 ^b^	0.47	0.047	0.029	0.022
Propionate, %	25.96 ^b^	27.49 ^ab^	27.78 ^a^	27.67 ^ab^	0.43	0.023	0.058	0.251
Butyrate, %	14.47 ^a^	11.99 ^c^	13.26 ^b^	14.27 ^ab^	0.57	0.158	<0.001	<0.001
Iso-butyrate, %	1.18 ^a^	1.02 ^b^	1.01 ^b^	1.01 ^b^	0.03	<0.001	0.003	0.024
Valerate, %	2.17 ^a^	1.61 ^b^	1.80 ^ab^	2.09 ^a^	0.13	0.560	0.001	0.015
Iso-valerate, %	2.69 ^a^	2.16 ^b^	2.16 ^b^	2.09 ^b^	0.14	<0.001	0.004	0.005
A/P ratio	2.06	2.03	1.94	1.92	0.03	0.017	0.594	0.822
IVDMD, %	72.82 ^a^	70.32 ^ab^	69.45 ^b^	69.64 ^b^	0.13	0.023	0.088	0.430

^a–c^ Means without a common superscript letter differ significantly (*p* < 0.05). L, Q, and, C mean linear, quadratic, and cubic patterns, respectively. ^1^ DM = dry matter; dDM, digestible dry matter; VFAs, volatile fatty acids; A/P ratio, acetate/propionate ratio; IVDMD, in vitro dry matter digestibility. ^2^ Substrates containing 0%, 1%, 3%, and 5% of *Rheum palmatum* root on a dry matter basis.

**Table 6 animals-14-02637-t006:** Effects of *Rheum palmatum* root addition on dry matter intake and methane emissions.

Variable	Treatments		SEM	*p*-Value
	CON	RP		
Dry matter intake, kg/day	5.76	5.33	0.367	0.073
Methane emission, L/day	221.1	216.1	54.10	0.914
Methane emission, g/day	158.4	154.8	38.75	0.914
Methane yield, g/d/kg DMI	24.40	25.11	5.650	0.885

CON, no additives; RP, concentrates containing 3% of *Rheum palmatum* root on a dry matter basis.

**Table 7 animals-14-02637-t007:** Effects of *Rheum palmatum* root addition on rumen fermentation characteristics.

Variable	Treatments		SEM	*p*-Value
	CON	RP		
pH	6.56	6.24	0.171	0.130
Ammonia-N, mg/dL	9.10	8.22	0.842	0.269
Total volatile fatty acids, m*M*	72.99	71.79	5.953	0.937
Acetate, %	67.55 ^b^	71.56 ^a^	1.062	0.023
Propionate, %	22.19	17.51	1.230	0.260
Butyrate, %	13.19	12.51	1.609	0.677
Iso-butyrate, %	1.62 ^a^	0.88 ^b^	0.180	0.021
Valerate, %	0.69	0.54	0.142	0.329
Iso-valerate, %	2.15	1.16	0.348	0.052
Acetate/propionate	3.04	4.09	0.542	0.164

CON, no additives; RP, concentrates containing 3% of *Rheum palmatum* root on a dry matter basis. ^a, b^ Means without a common superscript letter differ significantly (*p* < 0.05).

**Table 8 animals-14-02637-t008:** Effects of *Rheum palmatum* root addition on the blood metabolite levels.

Variable	Treatments		SEM	*p*-Value
	CON	RP		
Glucose, mg/dL	82.3	77.0	5.715	0.317
Creatine, mg/dL	1.17	1.07	0.153	0.468
Blood urea nitrogen, mg/dL	14.3	12.3	1.528	0.184
BUN/CREA	12.3	11.7	1.354	0.579
Inorganic phosphate, mg/dL	7.87	7.03	0.677	0.206
Calcium, mg/dL	9.37	9.07	0.372	0.379
Total protein, g/dL	6.60	6.57	0.376	0.919
Albumin, g/dL	2.93	3.07	0.183	0.422
Globulin, g/dL	3.67	3.50	0.337	0.616
Albumin/globulin	0.80	0.87	0.129	0.561
Alanine aminotransferase, U/L	46.7	46.0	6.298	0.903
Alkaline phosphatase, U/L	81.3	82.7	17.76	0.931
Gamma-glutamyl transferase, U/L	13.7	11.3	7.303	0.716
Total bilirubin, mg/dL	0.10	0.17	0.041	0.116
Cholesterol, mg/dL	95.0	70.3	25.46	0.301
Amylase, U/L	12.7	16.3	3.979	0.322
Lipase, U/L	36.7 ^b^	57.0 ^a^	7.461	0.029

CON, no additives; RP, concentrates containing 3% of *Rheum palmatum* root on a dry matter basis; BUN/CREA, blood urea nitrogen to creatinine ratio; albumin/globulin, albumin to globulin ratio. ^a, b^ Means without a common superscript letter differ significantly (*p* < 0.05).

## Data Availability

The data presented in this study are available upon request from the first author. The data are not publicly available because of restrictions imposed by the research group.

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
