# Peer review of "Effects of Rheum palmatum Root on In Vitro and In Vivo Methane Production and Rumen Fermentation Characteristics"

_animals, 2024, doi:10.3390/ani14182637_

Round 1

Reviewer 1 Report

Comments and Suggestions for Authors
  • Summary:  The goal of the study is to investigate the effects of the inclusion of Rheum palmatum (RP) on in vitro fermentation and methane production and in vivo rumen fermentation and blood metabolites in cattle.  
  • General concept comments
    • Strengths: The manuscript has clear hypothesis, and relevant content that has not been well explored in the literature (RP inclusion for cattle).
    • Weaknesses: There are some inconsistencies related to the methodology. The authors did not correlate the findings of both in vitro and in vivo experiments. Lacking connections on the Discussion section, 
  • Specific comments: 

Title: Delete “Korean cattle”. Use production instead of reduction.

Line 14: “The effects on methane production”.

Line 18: For how long the animals was fed with RP for the in vivo trial? Was the adaptation period enough? If the authors are not sure if the adaptation period was enough, there is a lack on methodology.

Line 36-44: This sentence should be deleted – this is guideline for writing.

Line 50: Replace materials with strategies.

Line 60: Replace medicinal crops with medicinal herbs.

Line 67: Delete “(also known as rhubarb)”.

Line 67 – 69: Citation?

Line 73: Missing citation of in vivo studies using Rheum palmatum.

Line 110: Chemical and phytochemical composition.

Table 1. I’m confused about the numbers in the phytochemicals (1 and 2). What do they mean?

Line 116: Missing adaptation period, feed samples collection protocol, etc.

Line 136: Were the substrates weighed in bags or directly into the flasks? Not clear how the incubation was performed.

Table 3: High CP, NDF and NDA for RP diet.  Is there a list of ingredients for each diet, control and RP? Is the table the diet offered or what the animals consumed?

Line 259: I would change RP supplementation to just RP. My understanding is that the RP was only added in the substrate. The ruminal liquid donors were not supplemented with RP.

Line 265: Add “with increasing level of RP”.

Line 266: Delete supplementation.

Table 4: Add footer for L, Q and C. Same for table 5.

Line 288: Delete Hanwoo cattle.

Line 290: DMI tended to decrease, the PR diet has greater % of fiber. Could it be a reason for reduced DWI?

Line 297: Delete Hanwoo cattle.

Table 6, 7, and 8: Delete Hanwoo cattle from the title. I don’t understand the focus on Hanwoo cattle. Shouldn’t the results apply to other cattle as well?

Line 309: Delete Hanwoo cattle.

Line 310 – 316: No difference between groups for the majority of the blood metabolite. No need to present all results since they are already presented in the table 8.

Line 334: Linear decrease in methane with increased levels of RP.

Line 338: “Who found a decrease in DM…”

Line 362-364: Can you say that the decrease in DMI was due to palatability? How about fiber intake? The diet did not have the same amount of protein, fiber and energy, so it makes it hard to compare them.

Line 369: How long does it take to the rumen bacteria to adapt to a new diet? Do you expect different results if the adaptation period was longer (21, 28 days)?

Line 372: This result was not presented. Did you do feed behavior? Or separation of leftovers?

Comments on the Quality of English Language

English can be improved. 

Author Response

Comments and Suggestions for Authors

  • Summary: The goal of the study is to investigate the effects of the inclusion of Rheum palmatum (RP) on in vitro fermentation and methane production and in vivo rumen fermentation and blood metabolites in cattle.  
  • General concept comments
    • Strengths: The manuscript has clear hypothesis, and relevant content that has not been well explored in the literature (RP inclusion for cattle).
    • Weaknesses: There are some inconsistencies related to the methodology. The authors did not correlate the findings of both in vitro and in vivo experiments. Lacking connections on the Discussion section, 
  • Specific comments:

Title: Delete “Korean cattle”. Use production instead of reduction.

Answer: Edited it.

Line 14: “The effects on methane production”.

Answer: Edited it.

Line 18: For how long the animals was fed with RP for the in vivo trial? Was the adaptation period enough? If the authors are not sure if the adaptation period was enough, there is a lack on methodology.

Answer: The adaptation period lasted 14 days, which is sufficient for cows. The reference below explains why an acclimation period should last 14 days.

  • Machado, M. G., Detmann, E., Mantovani, H. C., Valadares Filho, S. C., Bento, C. B., Marcondes, M. I., & Assunção, A. S. (2016). Evaluation of the length of adaptation period for changeover and crossover nutritional experiments with cattle fed tropical forage-based diets. Animal Feed Science and Technology222, 132-148.)

Line 36-44: This sentence should be deleted – this is guideline for writing.

Answer: Deleted it.

Line 50: Replace materials with strategies.

Answer: Replaced it.

Line 60: Replace medicinal crops with medicinal herbs.

Answer: Replaced it.

Line 67: Delete “(also known as rhubarb)”.

Answer: Deleted it.

Line 67 – 69: Citation?

Answer: Added a reference.

Line 73: Missing citation of in vivo studies using Rheum palmatum.

Answer: The sentence was edited.

Line 110: Chemical and phytochemical composition.

Answer: Edited it.

Table 1. I’m confused about the numbers in the phytochemicals (1 and 2). What do they mean?

Answer: The meaning of 1 or 2 for each compound is expressed when the positions at which certain chemical structures are attached are different.

Line 116: Missing adaptation period, feed samples collection protocol, etc.

Answer: The sentence presents in vitro trial not in vivo trial. The information was presented in L188-205.

Line 136: Were the substrates weighed in bags or directly into the flasks? Not clear how the incubation was performed.

Answer: The sentence was edited.

Table 3: High CP, NDF and NDA for RP diet.  Is there a list of ingredients for each diet, control and RP? Is the table the diet offered or what the animals consumed?

Answer: It is the table for the chemical composition of the offered diet (some contents were edited).

Line 259: I would change RP supplementation to just RP. My understanding is that the RP was only added in the substrate. The ruminal liquid donors were not supplemented with RP.

Answer: RP was supplemented in the Ankom bag with the substrate, so the sentence was not edited.

Line 265: Add “with increasing level of RP”.

Answer: Added it.

Line 266: Delete supplementation.

Answer: RP was supplemented in the Ankom bag with the substrate, so the sentence was not edited.

Table 4: Add footer for L, Q and C. Same for table 5.

Answer: Added it.

Line 288: Delete Hanwoo cattle.

Answer: Deleted it.

Line 290: DMI tended to decrease, the PR diet has greater % of fiber. Could it be a reason for reduced DWI?

Answer: The fiber value was changed. In my opinion, it is predicted that DMI was reduced in the RP diet due to the unique smell of herbal medicine.

Line 297: Delete Hanwoo cattle.

Answer: Deleted it.

Table 6, 7, and 8: Delete Hanwoo cattle from the title. I don’t understand the focus on Hanwoo cattle. Shouldn’t the results apply to other cattle as well?

Answer: Deleted it.

Line 309: Delete Hanwoo cattle.

Answer: Deleted it.

Line 310 – 316: No difference between groups for the majority of the blood metabolite. No need to present all results since they are already presented in the table 8.

Answer: The results were no different from the control group, but the absence of negative health effects from RP supplementation is meaningful. Therefore, this sentence was not modified

Line 334: Linear decrease in methane with increased levels of RP.

Answer: Edited it.

Line 338: “Who found a decrease in DM…”

Answer: The sentence was edited.

Line 362-364: Can you say that the decrease in DMI was due to palatability? How about fiber intake? The diet did not have the same amount of protein, fiber and energy, so it makes it hard to compare them.

Answer: In a preliminary experiment, PR powder was provided as a top dressing, but the cattle consumed the feed while picking out the PR powder. I confirmed the low palatability of PR powder in the preliminary experiment.

Line 369: How long does it take to the rumen bacteria to adapt to a new diet? Do you expect different results if the adaptation period was longer (21, 28 days)?

Answer: It is different depending on the study but several manuscripts reported that adaptation should be conducted for 14 days at least. In my opinion, the rumen bacteria could adapt to the diet.

  • Machado, M. G., Detmann, E., Mantovani, H. C., Valadares Filho, S. C., Bento, C. B., Marcondes, M. I., & Assunção, A. S. (2016). Evaluation of the length of adaptation period for changeover and crossover nutritional experiments with cattle fed tropical forage-based diets. Animal Feed Science and Technology222, 132-148.)

Line 372: This result was not presented. Did you do feed behavior? Or separation of leftovers?

 Answer: The result was not presented, but I confirmed that the cow separated the feed.

Reviewer 2 Report

Comments and Suggestions for Authors

Dear Editor and Authors,

The manuscript entitled “Effects of Rheum palmatum on in vitro and in vivo methane reduction and rumen fermentation characteristics in Korean cattle” by Lee et al., investigates an interesting topic to the field of ruminant nutrition, however some issues need to be addressed before further consideration. Please see my comments below.

Comments:

Simple Summary

L 13: To my understanding you used the root of Rheum palmatum. If yes, please specify that you are referring to the root, not the whole plant. Make this change throughout the manuscript.

L 14: It seems that the first part of the sentence refers to the in vitro and the other to the in vivo. Please clarify.

Abstract

L 20-21: Please include the design of in vitro trial.

L 27: If this refers to DMI from Table 6, then it is a tendency not an effect. Please specify that (i.e., tended to be lower in the RP…).

L 27-28: Using “significant” or “not significant” is redundant. When a variable is not significant it is better to state it as “no effect”. Please rephrase accordingly.

L 29: To improve the consistency of your manuscript please be consistent about the way you describe your results. Please specify the p-value.

Keywords

Please arrange them in alphabetical order.

Introduction

L 36-44: Remove. Please make sure to remove all the editorial/instruction to author comments.

L 44-47: Subjective statement that is quite generic. Please remove.

L 51-55: Lacks clarity. Please rephrase.

L 56-63: This paragraph needs some improvement. Please elaborate more on specific medicinal herbs you are referring to.

L 60: Add citation.

L 65: Add citation.

L 70-76: The main topic of these sentences is the limited number of in vivo trials evaluating the potential of RP. No need to make them so long. Please make them shorter and simple.

Material and Methods

L 138: Replace “fifty” with “50”.

L 141: Please remove “(control)”. 0% is self-explanatory.

L 146: Did you use separate bottles for 24-h and 48-h incubations or the same ones? Did you agitate the bottles during the incubation? Please clarify.

L 117 & 199: What’s the age of these animals?

L 199-200: Was there an adaptation period? Was there a washout period between the experimental periods? Did you collect samples before the start of the experimental diets?

L 210: 2 weeks before the start of the experimental period?

L 232: Remove “s” from “fluids”.

L 237-243: Did you analyze as repeated measures?

L 244-249: Did you evaluate the normality of residuals (if yes, how?), and homogeneity of variance? Why did you remove period from the model? Please explain.

L 244-249: Covariates (if any) were included to the model?

Results

General comments

Results should be discussed in the order that are presented in the tables, if they are significant. For example, on Table 4, the first variable you detected significance was the CH4 (mL/ g of dDM; p = 0.036). This variable should be discussed first in your results section etc. After you should mention the variables that you found no effects. To improve the clarity and consistency of your manuscript please consider adjusting that.

Specific comments

L 259: Add “The” before “RP”.

L 264 & 301: “iso-butyrate”. Please correct.

L 292: Remove “in”.

Discussion

L 342 & 349: Rheum spp.

Conclusion

L 389: Rheum palmatum ? Also, make sure to state that you are referring to the root not the whole plant.

Tables: Table 6: Typo at RP.

Comments on the Quality of English Language

Please refer to the Comments and Suggestions for Authors.

Author Response

Comments and Suggestions for Authors

Dear Editor and Authors,

The manuscript entitled “Effects of Rheum palmatum on in vitro and in vivo methane reduction and rumen fermentation characteristics in Korean cattle” by Lee et al., investigates an interesting topic to the field of ruminant nutrition, however some issues need to be addressed before further consideration. Please see my comments below.

Comments:

Simple Summary

L 13: To my understanding you used the root of Rheum palmatum. If yes, please specify that you are referring to the root, not the whole plant. Make this change throughout the manuscript.

Answer: Changed all.

L 14: It seems that the first part of the sentence refers to the in vitro and the other to the in vivo. Please clarify.

Abstract: Each trial was presented as "in the in vitro trial" or "in the in vivo trial".

L 20-21: Please include the design of in vitro trial.

Answer: included the design of in vitro trial.

L 27: If this refers to DMI from Table 6, then it is a tendency not an effect. Please specify that (i.e., tended to be lower in the RP…).

Answer: Edited it.

L 27-28: Using “significant” or “not significant” is redundant. When a variable is not significant it is better to state it as “no effect”. Please rephrase accordingly.

Answer: Edited it.

L 29: To improve the consistency of your manuscript please be consistent about the way you describe your results. Please specify the p-value.

Answer: All p-values were deleted in the abstract due to the word limit (maximum 200)

Keywords

Please arrange them in alphabetical order.

Answer: Arranged all.

Introduction

L 36-44: Remove. Please make sure to remove all the editorial/instruction to author comments.

Answer: Removed the sentence.

L 44-47: Subjective statement that is quite generic. Please remove.

Answer: I believe this is not a very general subjective statement, and I have added supporting references for it.

L 51-55: Lacks clarity. Please rephrase.

Answer: In my opinion, there is no need to add more content to lines 51-55, as it is not important and would only add unnecessary references. Information about the medicinal plant is provided in the following sentence.

L 56-63: This paragraph needs some improvement. Please elaborate more on specific medicinal herbs you are referring to.

Answer: Information on medicinal herbs other than Rheum palmatum is not important, so this paper was not revised. Information about Rheum palmatum is mentioned starting in line 64.

L 60: Add citation.

Answer: Added reference.

L 65: Add citation.

Answer: Added reference.

L 70-76: The main topic of these sentences is the limited number of in vivo trials evaluating the potential of RP. No need to make them so long. Please make them shorter and simple.

Answer: made the sentences shorter and simpler. (deleted [9])

Material and Methods

L 138: Replace “fifty” with “50”.

Answer: Replaced it.

L 141: Please remove “(control)”. 0% is self-explanatory.

Answer: Removed it

L 146: Did you use separate bottles for 24-h and 48-h incubations or the same ones? Did you agitate the bottles during the incubation? Please clarify.

Answer: The information was added.

L 117 & 199: What’s the age of these animals?

Answer: The information was added.

L 199-200: Was there an adaptation period? Was there a washout period between the experimental periods? Did you collect samples before the start of the experimental diets?

Answer: The information was mentioned in L200-201 and L205-206.

L 210: 2 weeks before the start of the experimental period?

Answer: The measurement period was conducted after 14 days (adaptation period).

L 232: Remove “s” from “fluids”.

Answer: Deleted it.

L 237-243: Did you analyze as repeated measures?

Answer: The in vitro trial was performed by independently separating rumen fluid from each cannulated cow.

L 244-249: Did you evaluate the normality of residuals (if yes, how?), and homogeneity of variance? Why did you remove period from the model? Please explain.

Answer: Normality was evaluated by using Proc GLIMMIX and PROC univariate (The test was conducted on Shapiro-Wilk, Cramer-von Mises, and Anderson-Darling). And I added period in the sentsence.

L 244-249: Covariates (if any) were included to the model?

Answer: No additions were made other than those presented in this study.

Results

General comments

Results should be discussed in the order that are presented in the tables, if they are significant. For example, on Table 4, the first variable you detected significance was the CH4 (mL/ g of dDM; p = 0.036). This variable should be discussed first in your results section etc. After you should mention the variables that you found no effects. To improve the clarity and consistency of your manuscript please consider adjusting that.

Answer: Added it.

Specific comments

L 259: Add “The” before “RP”.

Answer: Added it.

L 264 & 301: “iso-butyrate”. Please correct.

Answer: Edited it.

L 292: Remove “in”.

Answer: Deleted "in".

Discussion

L 342 & 349: Rheum spp.

Answer: Edited it.

Conclusion

L 389: Rheum palmatum ? Also, make sure to state that you are referring to the root not the whole plant.

Answer: Edited it.

Tables: Table 6: Typo at RP.

Answer: Edited it.

Reviewer 3 Report

Comments and Suggestions for Authors

This manuscript reports the effect of Rheum palmatum root (rhubarb) on in vitro and in vivo methane production and ruminal fermentation. The study is not highly novel as rhubarb has been evaluated in some studies. 

Here are some suggestions and comments:

L137: was the buffer prepared anaerobically? Or the headspace was only flushed with CO2?

L147: the symbols (Ø) are not clear to me.

L199: The number of animals were low for this type of study.

L216: is the washout period besides the usual 21 days experimental period?

Table 3: what was the ingredient composition of the concentrates? How could CP content in RP diet be higher than the control when the root of RP had lower protein (Table 1)? Also, how could NDF content be greater in RP diet when RP had low NDF concentration?

Table 5: IVDMD of 1% and 3% was lower, but VFA concentration was substantially greater - what was the reason?

L342: I think you meant to here such as fibrolytic bacteria instead of such as fibrinolytic bacteria.

L372-373: Please report the concentrate and forage intake in the table 6 to understand the explanations.

L376-377: But in this study ammonia concentration was similar - how do you explain it in your study situation?

L377-378: "Wang et al. [29] reported that increased plant extract levels 377
led to a decrease in CP and branched-chain fatty acids." This sentence is unclear to me.

Comments on the Quality of English Language

Acceptable.

Author Response

This manuscript reports the effect of Rheum palmatum root (rhubarb) on in vitro and in vivo methane production and ruminal fermentation. The study is not highly novel as rhubarb has been evaluated in some studies. 

Here are some suggestions and comments:

L137: was the buffer prepared anaerobically? Or the headspace was only flushed with CO2?

Answer: the buffered rumen fluid was flushed with CO2 on the headspace (edited the information in this manuscript.)

L147: the symbols (Ø) are not clear to me.

Answer: Edited the symbols. Please let me know again if the symbols are not clear to you.

L199: The number of animals were low for this type of study.

Answer: I agree with your opinion, but the animal experiment is not completely impossible if the experiment is conducted with cattle under similar conditions, Several studies have also been performed. (The references below are studies with similar experimental designs and numbers of animals.)

  • Gadulrab, K.et al. (2023). Effect of Feeding Dried Apple Pomace on Ruminal Fermentation, Methane Emission, and Biohydrogenation of Unsaturated Fatty Acids in Dairy Cows. Agriculture13(10), 2032.
  • Ma, Z. Y. et al. (2019). Molecular hydrogen produced by elemental magnesium inhibits rumen fermentation and enhances methanogenesis in dairy cows. Journal of dairy science102(6), 5566-5576.
  • Petersen, S. O. et al. (2015). Dietary nitrate for methane mitigation leads to nitrous oxide emissions from dairy cows. Journal of environmental quality44(4), 1063-1070.

L216: is the washout period besides the usual 21 days experimental period?

Answer: Yes, the washout period was conducted for 14 days independently with 21-days experimental period (edited the sentence).

Table 3: what was the ingredient composition of the concentrates? How could CP content in RP diet be higher than the control when the root of RP had lower protein (Table 1)? Also, how could NDF content be greater in RP diet when RP had low NDF concentration?

Answer: The ingredient composition of the concentrates is same with Table 1. Chemical composition of Table 3.

Table 5: IVDMD of 1% and 3% was lower, but VFA concentration was substantially greater - what was the reason?

Answer: An explanation may be possible if microbial properties or metabolites were analyzed, but it is difficult to explain with the current limited data.

L342: I think you meant to here such as fibrolytic bacteria instead of such as fibrinolytic bacteria.

Answer: Edited it.

L372-373: Please report the concentrate and forage intake in the table 6 to understand the explanations.

Answer: The concentrate and forage were mixed when cows were fed the diet. Therefore, it cannot measure the individual feed intake of concentrate and forage.

L376-377: But in this study ammonia concentration was similar - how do you explain it in your study situation?

Answer: The sentence explains on the decrease of branched-chain fatty acids (iso-butyrate; p=0.023 and iso-valerate; p=0.052) in the present study.

L377-378: "Wang et al. [29] reported that increased plant extract levels 377
led to a decrease in CP and branched-chain fatty acids." This sentence is unclear to me.

Answer: In the present study, CP digestibility was not analyzed but demonstrated the decrease of branched-chain fatty acids (iso-butyrate; p=0.023 and iso-valerate; p=0.052).

Round 2

Reviewer 1 Report

Comments and Suggestions for Authors

Comments and Suggestions for Authors 

  • The authors have addressed some of the inconsistencies related to the methodology. I'm still missing a connection between the in vitro and in vivo results in the discussion section. 

  • The authors keep discussing and concluding that the discrepancy of in vitro and in vivo may be attributed to the adaptation of ruminal bacteria to RP. If this is the case, the methodology was not correct.

  • It seems like the authors did not take responding to the reviewer's comments and questions very seriously. 

Line 62: "Several studies have demonstrated the effects of Rheum palmatum on the improvement of rumen fermentation in an in vitro batch system..." 

Where are the citations? There is no citation for this sentence. 

Table 3: The table was edited for the control diet. What changed from submission to review? 

Table 3: it is the first time I’m reading Italian ryegrass. If the forage used was Italian ryegrass you need to say that. Was it offered fresh, hay, etc?  

Line 255: I still don’t think that supplementation is the right word. You added RP in the bag. Change to “the addition of RP” or only “RP”.  

Line 262: same as above. Change the word supplementation. 

Line 305 – 309: If you want to highlight that there were no negative health effects of RP supplementation do it in the discussion section. My point was: you don’t have to present all the results that were already presented in the table. Delete this part, you can combine in one sentence like there were no differences in blood metabolites between treatments, with the exception of....  

Line 357: this is an awkwardly worded sentence. Delete either supplementation or mixed. 

Line 358: Can you say that the decrease in DMI was due to palatability? How about fiber intake? The diets did not have the same amount of protein, fiber and energy, so it makes it hard to compare them. 

The author’s answer: In a preliminary experiment, PR powder was provided as a top dressing, but the cattle consumed the feed while picking out the PR powder. I confirmed the low palatability of PR powder in the preliminary experiment.  

Please cite this study then.  

Line 367: this is an awkwardly worded sentence. If the animals picked concentrate, how did they eat more forage? 

Authors' answer: The result was not presented, but I confirmed that the cow separated the feed.  

Again, just your confirmation that they separated concentrate and forage isn’t enough for a scientific publication.

Comments on the Quality of English Language

There are still some sentences awkwardly worded.

Author Response

Comments and Suggestions for Authors

  • The authors have addressed some of the inconsistencies related to the methodology. I'm still missing a connection between the in vitro and in vivo results in the discussion section. 
  • The authors keep discussing and concluding that the discrepancy of in vitro and in vivo may be attributed to the adaptation of ruminal bacteria to RP. If this is the case, the methodology was not correct.
  • It seems like the authors did not take responding to the reviewer's comments and questions very seriously. 

Line 62: "Several studies have demonstrated the effects of Rheum palmatum on the improvement of rumen fermentation in an in vitro batch system..." 

Where are the citations? There is no citation for this sentence. 

Answer: added reference.

Table 3: The table was edited for the control diet. What changed from submission to review? 

Answer: There was an error in the process of copying previously used data, and we missed the check it.

Table 3: it is the first time I’m reading Italian ryegrass. If the forage used was Italian ryegrass you need to say that. Was it offered fresh, hay, etc?  

Answer: I offered IRG hay and the information was added in Table 3.

Line 255: I still don’t think that supplementation is the right word. You added RP in the bag. Change to “the addition of RP” or only “RP”.  

Answer: I changed the word.

Line 262: same as above. Change the word supplementation. 

Answer: I changed the word.

Line 305 – 309: If you want to highlight that there were no negative health effects of RP supplementation do it in the discussion section. My point was: you don’t have to present all the results that were already presented in the table. Delete this part, you can combine in one sentence like “ there were no differences in blood metabolites between treatments, with the exception of...”.  

Answer: edited it.

Line 357: this is an awkwardly worded sentence. Delete either supplementation or mixed. 

Answer: edited it.

Line 358: Can you say that the decrease in DMI was due to palatability? How about fiber intake? The diets did not have the same amount of protein, fiber and energy, so it makes it hard to compare them. 

The author’s answer: In a preliminary experiment, PR powder was provided as a top dressing, but the cattle consumed the feed while picking out the PR powder. I confirmed the low palatability of PR powder in the preliminary experiment.  

Please cite this study then.  

Answer: It was visually inspected but not weighed. However, it is considered to be sufficiently predictable because the reference (Akbar, 2020) reported that there was a part where the palatability was low.

Line 367: this is an awkwardly worded sentence. If the animals picked concentrate, how did they eat more forage? 

Authors' answer: The result was not presented, but I confirmed that the cow separated the feed.  

Again, just your confirmation that they separated concentrate and forage isn’t enough for a scientific publication.

Answer: I have no data, so the sentence was deleted.

Reviewer 2 Report

Comments and Suggestions for Authors

Dear Authors,

To further improve the quality of your manuscript please replace "milliliters" with "mL" on L131.

Author Response

To further improve the quality of your manuscript please replace "milliliters" with "mL" on L131.

Answer: Edited it.

Reviewer 3 Report

Comments and Suggestions for Authors

Authors revised some suggestions, but many comments are not addressed in the paper. Authors should revise it in the paper.

Here are my comments:

Table 3: what was the ingredient composition of the concentrates? How could CP content in RP diet be higher than the control when the root of RP had lower protein (Table 1)? Also, how could NDF content be greater in RP diet when RP had low NDF concentration?

Answer: The ingredient composition of the concentrates is same with Table 1. Chemical composition of Table 3.

Comments: The explanations are not understandable. Table 1 report composition of the RP root. Authors specifically answer the question.

Table 5: IVDMD of 1% and 3% was lower, but VFA concentration was substantially greater - what was the reason?

Answer: An explanation may be possible if microbial properties or metabolites were analyzed, but it is difficult to explain with the current limited data.

Comment: Authors should attempt to explain the results. As a reader, I should know it in the paper. Authors must check the VFA data or remove this type of data if they are unable to explain.

L372-373: Please report the concentrate and forage intake in the table 6 to understand the explanations.

Answer: The concentrate and forage were mixed when cows were fed the diet. Therefore, it cannot measure the individual feed intake of concentrate and forage.

Comments: Then how did you say "In this study, the RP group picked concentrate and fed as much forage as possible. This means that the RP group had a higher forage diet than the CON group, which may have led to an increase in acetate concentration".

L376-377: But in this study ammonia concentration was similar - how do you explain it in your study situation?

Answer: The sentence explains on the decrease of branched-chain fatty acids (iso-butyrate; p=0.023 and iso-valerate; p=0.052) in the present study.

comment: If BCFA is produced from amino acid deamination, then ammonia should be higher, but it is not case. Please explain for the readers. Scientific explanations are not only based one direct variable, but it should related other variables in the study context.

L377-378: "Wang et al. [29] reported that increased plant extract levels
led to a decrease in CP and branched-chain fatty acids." This sentence is unclear to me.

Answer: In the present study, CP digestibility was not analyzed but demonstrated the decrease of branched-chain fatty acids (iso-butyrate; p=0.023 and iso-valerate; p=0.052).

Comment: Here I did not ask explanation, but the sentence expression is unclear. Still the sentence is unclear to me "Yang et al. [32] reported that increased plant extract levels led to a decrease in CP and branched-chain fatty acids." Can the authors themselves understand this sentence.

Overall the authors responses to the comments is not satisfactory for the discussion. Authors must try to improve the discussions and explain all the significant results.

This is not a novel study, nor the animal design is highly sound (low number and short duration). Reviewers should then expect better scientific presentation.

Comments on the Quality of English Language

Acceptable.

Author Response

Comments and Suggestions for Authors

Authors revised some suggestions, but many comments are not addressed in the paper. Authors should revise it in the paper.

Here are my comments:

Table 3: what was the ingredient composition of the concentrates? How could CP content in RP diet be higher than the control when the root of RP had lower protein (Table 1)? Also, how could NDF content be greater in RP diet when RP had low NDF concentration?

Answer: The ingredient composition of the concentrates is same with Table 1. Chemical composition of Table 3.

Comments: The explanations are not understandable. Table 1 report composition of the RP root. Authors specifically answer the question.

Answer: In Table 3, there was an error in the values for the control group, which has now been corrected. Upon reviewing the revised data, it was confirmed that the CP content and NDF content of the control group were higher than those in the RP diet. I apologize for any confusion this may have caused.

Table 5: IVDMD of 1% and 3% was lower, but VFA concentration was substantially greater - what was the reason?

Answer: An explanation may be possible if microbial properties or metabolites were analyzed, but it is difficult to explain with the current limited data.

Comment: Authors should attempt to explain the results. As a reader, I should know it in the paper. Authors must check the VFA data or remove this type of data if they are unable to explain.

 Answer: Added discussion comment about the results. (L351-)

L372-373: Please report the concentrate and forage intake in the table 6 to understand the explanations.

Answer: The concentrate and forage were mixed when cows were fed the diet. Therefore, it cannot measure the individual feed intake of concentrate and forage.

Comments: Then how did you say "In this study, the RP group picked concentrate and fed as much forage as possible. This means that the RP group had a higher forage diet than the CON group, which may have led to an increase in acetate concentration".

 Answer: L370-372 was deleted.

L376-377: But in this study ammonia concentration was similar - how do you explain it in your study situation?

Answer: The sentence explains on the decrease of branched-chain fatty acids (iso-butyrate; p=0.023 and iso-valerate; p=0.052) in the present study.

comment: If BCFA is produced from amino acid deamination, then ammonia should be higher, but it is not case. Please explain for the readers. Scientific explanations are not only based one direct variable, but it should related other variables in the study context.

Answer: The sentence was edited.

L377-378: "Wang et al. [29] reported that increased plant extract levels
led to a decrease in CP and branched-chain fatty acids." This sentence is unclear to me.

Answer: In the present study, CP digestibility was not analyzed but demonstrated the decrease of branched-chain fatty acids (iso-butyrate; p=0.023 and iso-valerate; p=0.052).

Comment: Here I did not ask explanation, but the sentence expression is unclear. Still the sentence is unclear to me "Yang et al. [32] reported that increased plant extract levels led to a decrease in CP and branched-chain fatty acids." Can the authors themselves understand this sentence.

 Answer: The sentence was edited.

Overall the authors responses to the comments is not satisfactory for the discussion. Authors must try to improve the discussions and explain all the significant results.

This is not a novel study, nor the animal design is highly sound (low number and short duration). Reviewers should then expect better scientific presentation.

 Answer: Thank you for your thoughtful comments. I have revised the content based on your comments, and if you have any additional comments, please let me know.

Round 3

Reviewer 1 Report

Comments and Suggestions for Authors

The abstract section can be improved. See my changes below:

Abstract: This study investigated the impact of Rheum palmatum root (RP) on rumen fermentation, blood metabolites and methane production in cattle. For the in vitro trial, rumen fluid was collected from three cannulated Korean native steers (736 ± 15 kg) and mixed with McDougal buffer (1:3 ratio). Treatments were divided into control and RP supplement groups (1%, 3%, and 5% of substrates), with each sample incubated at 39°C for 24 and 48 hours. In vivo trials involved four Korean steers in a 2×2 crossover design over 3 26 weeks, with treatments including a control group and a group with 3% RP addition. In vitro methane production showed a dose-dependent linear decrease after 48 hours of incubation. Quadratic changes were observed in total volatile fatty acids, acetate, and butyrate. Additionally, in vitro dry matter digestibility decreased linearly with RP inclusion. From the in vivo trial, dry matter intake (DMI) tended to decrease in the RP group compared to the control. Methane emissions (g/kg DMI) had no effect by RP addition. Blood metabolites indicated higher lipase concentrations in the RP group. In conclusion, RP reduced methane production in the in vitro trial but had no effect in the in vivo trial, likely due to adaptation of ruminal bacteria to RP.

Author Response

Comments and Suggestions for Authors

The abstract section can be improved. See my changes below:

Abstract: This study investigated the impact of Rheum palmatum root (RP) on rumen fermentation, blood metabolites and methane production in cattle. For the in vitro trial, rumen fluid was collected from three cannulated Korean native steers (736 ± 15 kg) and mixed with McDougal buffer (1:3 ratio). Treatments were divided into control and RP supplement groups (1%, 3%, and 5% of substrates), with each sample incubated at 39°C for 24 and 48 hours. In vivo trials involved four Korean steers in a 2×2 crossover design over 3 26 weeks, with treatments including a control group and a group with 3% RP addition. In vitro methane production showed a dose-dependent linear decrease after 48 hours of incubation. Quadratic changes were observed in total volatile fatty acids, acetate, and butyrate. Additionally, in vitro dry matter digestibility decreased linearly with RP inclusion. From the in vivo trial, dry matter intake (DMI) tended to decrease in the RP group compared to the control. Methane emissions (g/kg DMI) had no effect by RP addition. Blood metabolites indicated higher lipase concentrations in the RP group. In conclusion, RP reduced methane production in the in vitro trial but had no effect in the in vivo trial, likely due to adaptation of ruminal bacteria to RP.

Answer: I edited some sentences depending on your opinion, but I can’t change all due to word limitation (maximum 200 words)

Reviewer 3 Report

Comments and Suggestions for Authors

Authors in this revision has tried to explain the results instead of rebuttals. This is appreciated.

Some minor comments

L349-354: What was the tannin concentration in RP that the amount will increase VFA concentration when fermented, while decreasing DMD? If I give example 1% RP addition will increase 1% x 5% tannin (if I assume) = 0.05%, but the increase of TVFA was too high though it decreased DMD. Authors should explain with some rationale.

Author should the values. I can assume there are some values that are not normal as I can see from the SEM. Remove the outliers and reanalysis.

L374-375: "They hypothesized that this was because ammonia could be formed from urea."

Comments: was in your study urea added? Please explain here.

Comments on the Quality of English Language

Acceptable.

Author Response

Comments and Suggestions for Authors

Authors in this revision has tried to explain the results instead of rebuttals. This is appreciated.

Some minor comments

L349-354: What was the tannin concentration in RP that the amount will increase VFA concentration when fermented, while decreasing DMD? If I give example 1% RP addition will increase 1% x 5% tannin (if I assume) = 0.05%, but the increase of TVFA was too high though it decreased DMD. Authors should explain with some rationale.

Author should the values. I can assume there are some values that are not normal as I can see from the SEM. Remove the outliers and reanalysis.

Answer: Unfortunately, I have no data for tannin concentration in PR. I checked the data to remove outliers, but further editing is difficult.

L374-375: "They hypothesized that this was because ammonia could be formed from urea."

Comments: was in your study urea added? Please explain here.

Answer: I didn't add urea separately.